

# 3D skeletal muscle fascicle engineering is improved with TGF-β1 treatment of myogenic cells and their co-culture with myofibroblasts

Jessica Krieger[1], Byung-Wook Park[2], Christopher R. Lambert[3] and Christopher Malcuit[1]

[1] Department of Biomedical Sciences, Kent State University, Kent, OH, United States of America
[2] Department of Civil/Environmental & Chemical Engineering, Youngstown State University, Youngstown, OH, United States of America
[3] Chemistry & Biochemistry, Worcester Polytechnic Institute, Worcester, MA, United States of America

Corresponding author
Jessica Krieger, jkrieger@kent.edu

## ABSTRACT

**Background**. Skeletal muscle wound healing is dependent on complex interactions between fibroblasts, myofibroblasts, myogenic cells, and cytokines, such as TGF-β1. This study sought to clarify the impact of TGF-β1 signaling on skeletal muscle cells and discern between the individual contributions of fibroblasts and myofibroblasts to myogenesis when in co-culture with myogenic cells. 3D tissue-engineered models were compared to equivalent 2D culture conditions to assess the efficacy of each culture model to predictively recapitulate the *in vivo* muscle environment.

**Methods**. TGF-β1 treatment and mono-/co-cultures containing human dermal fibroblasts or myofibroblasts and C2C12 mouse myoblasts were assessed in 2D and 3D environments. Three culture systems were compared: cell monolayers grown on 2D dishes and 3D tissues prepared via a self-assembly method or collagen 1-based hydrogel biofabrication. qPCR identified gene expression changes during fibroblast to myofibroblast and myoblast differentiation between culture conditions. Changes to cell phenotype and tissue morphology were characterized via immunostaining for myosin heavy chain, procollagen, and α-smooth muscle actin. Tissue elastic moduli were measured with parallel plate compression and atomic force microscopy systems, and a slack test was employed to quantify differences in tissue architecture and integrity.

**Results**. TGF-β1 treatment improved myogenesis in 3D mono- and co-cultures containing muscle cells, but not in 2D. The 3D TGF-β1-treated co-culture containing myoblasts and myofibroblasts expressed the highest levels of myogenin and collagen 1, demonstrating a greater capacity to drive myogenesis than fibroblasts or TGF-β1-treatment in monocultures containing only myoblasts. These constructs possessed the greatest tissue stability, integrity, and muscle fiber organization, as demonstrated by their rapid and sustained shortening velocity during slack tests, and the highest Young's modulus of 6.55 kPA, approximate half the stiffness of *in situ* muscle. Both self-assembled and hydrogel-based tissues yielded the most multinucleated, elongated, and aligned muscle fiber histology. In contrast, the equivalent 2D co-culture model treated with TGF-β1 completely lacked myotube formation through suppression of myogenin gene expression.

**Discussion**. These results show skeletal muscle regeneration can be promoted by treating myogenic cells with TGF-β1, and myofibroblasts are superior enhancers of myogenesis than fibroblasts. Critically, both TGF-β1 treatment and co-culturing skeletal muscle cells with myofibroblasts can serve as myogenesis accelerators across multiple tissue engineering platforms. Equivalent 2D culture systems cannot replicate these affects, however, highlighting a need to continually improve *in vitro* models for skeletal muscle development, discovery of therapeutics for muscle regeneration, and research and development of *in vitro* meat products.

# INTRODUCTION

Muscle regeneration occurs when muscle fibers are damaged during exercise or injury and quiescent satellite cells are activated to a proliferative myoblast phenotype (*Hill, Wernig & Goldspink, 2003*). Terminal differentiation is promoted through expression of the myogenic transcription factor myogenin (MYOG) and exit from the cell cycle. Cell fusion with an injured muscle fiber or other myoblasts forms a nascent syncytial myofiber (*Charge & Rudnicki, 2004*; *Le Grand & Rudnicki, 2007*). Fibroblasts support and stabilize muscle fiber architecture and biomechanics through basement membrane synthesis and facilitate muscle regeneration with extracellular matrix (ECM) deposition and remodeling (*Murphy et al., 2011*; *Sandbo & Dulin, 2011*; *Sanes, 2003*). The interaction between myoblasts and fibroblasts, two predominant cell types involved in skeletal muscle regeneration, with surrounding ECM and trophic factors determine healing outcomes. Transforming growth factor beta 1 (TGF-β1), produced by multiple cell types during wound healing, is a primary mediator of the mechanical, biochemical, and cellular behaviors observed in response of muscle to injury (*Karalaki et al., 2009*). TGF-β1 signaling differentiates fibroblasts into myofibroblasts, which are significant producers of collagen I (COL I), the main protein component of scar tissue, and α-smooth muscle actin (α-SMA), a highly contractile form of actin (*Mendias et al., 2012*). Normally these proteins are transiently expressed during muscle regeneration and contribute to tissue remodeling by temporarily providing physical substrates and biochemical cues for muscle fiber regeneration. However, excessive TGF-β1-mediated myofibroblast activation leads to ECM accumulation and tissue stiffening that decreases the ability of myoblasts to regenerate muscle (*Gilbert et al., 2010*; *Smith et al., 2011*).

The use of culture models consisting solely of myoblasts to investigate skeletal muscle regeneration is narrow in scope and validity and prevents effective screening of therapeutics. Using co-culture systems allows the bi-directional signaling between fibroblasts and myoblasts that attenuates and stabilizes myogenesis. Additionally, while 2D *in vitro* cell culture systems have demonstrable value, they are limited by a number of factors that have unwanted influence on cell behavior. After isolation, primary cells quickly lose their *in*

*situ* characteristics in response to a mechanically and biochemically alien environment (*Janson et al., 2013*). Fibroblasts and muscle stems cells are particularly sensitive to mechanical stimulation, and the rigidity of substrata can mask the cellular responses under investigation (*Engler et al., 2006*; *Godbout et al., 2013*). It is therefore highly valuable to further optimize the *in vitro* recapitulation of the *in vivo* environment. In this regard, myogenesis should be studied within the context of tissues rather than cell culture plates. Engineered tissue that contains aligned muscle fibers embedded within connective tissue is biomimetic to muscle fascicles observed *in vivo* (*Turrina, Martinez-Gonzalez & Stecco, 2013*; *Dennis et al., 2001*). The inclusion of fibroblasts and myofibroblasts is therefore our design target for modeling skeletal muscle. Although myofibroblasts are primarily known for their role in tissue regeneration, their inclusion in engineered skeletal muscle tissues has not been investigated, and the effectiveness of using myofibroblasts has not been compared to the known performance of fibroblasts.

Signaling of profibrotic factors such as TGF-β1 can deregulate the regenerative capacity of muscle and drive fibrosis when in excess (*Lieber & Ward, 2013*; *Mann et al., 2011*) but regularly contribute to muscle regeneration at moderate levels. The use of TGF-β1 may therefore be a means to organize tissue-engineered skeletal muscle development *in vitro*. A comprehensive and unifying characterization of the impact of TGF-β1 on myogenesis in different culture models has not been established, however. Consequently, this study sought to investigate the influence of TGF-β1 treatment on myogenesis in 2D and 3D culture models, and whether myofibroblasts outperform fibroblasts in accelerating myogenesis when co-cultured with myoblasts. 3D conditions included self-assembled, scaffoldless tissue constructs first developed by Gwyther (*Gwyther et al., 2011*; *Strobel et al., 2018*) and collagen 1-based hydrogels, while 2D models included cells grown on plastic culture plates. Monocultures consisted of murine C2C12 myoblasts, human dermal fibroblasts, or TGF-β1-differentiated myofibroblasts, and co-cultures were composed of myoblasts with either fibroblasts or myofibroblasts.

Our results show TGF-β1 signaling in 3D improves myogenesis in myoblast monocultures and co-cultures with myofibroblasts. In co-cultures, myofibroblasts enhanced muscle differentiation to a greater extent than fibroblasts. Our scaffoldless self-assembled tissues and hydrogels containing C2C12s both similarly displayed enhanced myogenesis with inclusion of myofibroblasts and TGF-β1 treatment, indicating this technique has wide applicability across a variety of 3D platforms. However, 2D experiments demonstrated TGF-β1-mediated inhibition of myogenesis in both mono- and co-culture conditions. Our data highlights that 2D *in vitro* models of skeletal muscle obscure a complete understanding of mechanisms of muscle regeneration, in addition to the usefulness of TGF-β1 treatment and inclusion of myofibroblasts in improving tissue engineered models.

## MATERIALS AND METHODS

### Cell and tissue culture

As shown in the experimental design listed in Fig. 1A, cultures consisting solely of human neo-natal dermal fibroblasts (passage 4 hDFs, PCS-201-010; ATCC, Manassas,

VA, USA) or mouse myoblasts (pre-passage 4 C2C12s, CRL-1772, ATCC) were plated on standard cell culture plates at 1 k/cm$^2$ in growth media (GM) consisting of DMEM (Gibco, Waltham, MA, USA), 10% fetal bovine serum (FBS; Hyclone, Pittsburgh, PA, USA), 1% penicillin/streptomycin (P/S, Gibco). To promote differentiation of fibroblasts to myofibroblasts, some fibroblasts were incubated with 1 ng/mL TGF-β1 (Peprotech, Rocky Hill, NJ, USA) for 6 days between P4 and P5. C2C12s and P4 fibroblasts and myofibroblasts were trypsinized, resuspended in GM, and seeded at a 1:1 ratio on standard cell culture plates at 20 k/cm$^2$ or self-assembled into 3D tissue constructs. To develop 3D self-assembled tissues, cell suspensions were added into custom 2% agarose molds that were prepared by casting molten agarose onto a patterned PDMS mask. After solidification of the gel, the molds were sectioned and added to cell culture plates. Each tissue mold is composed of a hollow well with a 2 mm diameter central agarose post (Fig. 1A). When a cell suspension is added to the well, the cells will prefer to adhere to each other instead of the agarose, and self-assemble into a ring-shaped tissue structure that contracts around the central post. A cell seeding density of 350,000 cells/millimeter of post diameter is used per tissue sample. This density was selected through a pilot experiment determining the concentration of cells required to self-assemble tissue around the circumference of the agarose post, which is a function of the post's diameter (C Malcuit, pers. comm., 2013). Hydrogels were produced from a protocol adapted from *Langelaan et al. (2011)* where $1.5 \times 10^6$ C2C12s alone or a 1:3 ratio of fibroblasts/myofibroblasts and C2C12s were resuspended in 42% GM, 54% rat collagen type 1 (3 mg/mL; Corning Life Sciences, Corning, NY, USA), and 2.7% NaOH (0.1 M). The hydrogel mixture was poured between house-shaped Velcro anchors super glued to the bottom of a six well plate and gelled for 45 min before adding GM. After 24 h, the media were switched in both 2D and 3D conditions to differentiation medium (DM) composed of DMEM, 2% horse serum (Sigma-Aldrich, St. Louis, MO, USA), and 1% P/S, with or without 1 ng/mL TGF-β1. Media was changed every 2–3 days for 7 days, after which cells and tissues underwent qPCR, ICC or IHC, or mechanical characterization. See Fig. 1B for listing of experimental groups.

## Gene expression analysis

Total RNA was isolated from cultures using TriZol reagent (Life Technologies, Carlsbad, CA, USA) according to the manufacturers recommended instructions. cDNA was reverse transcribed from 1 μg total RNA using qScript SuperMix (Quanta Biosciences, Gaithersburg, MD, USA) according to manufacturers instructions. qPCR was carried out using a 5 ng equivalent of cDNA in a 1X reaction of PerfeCTa SYBR Green SuperMix (Quanta Biosciences, Beverly, MA, USA) and 250 nM each (forward and reverse) custom oligonucleotide primers (Integrated DNA Technologies, Coralville, IA, USA) generated by using PrimerBlast (National Center for Biotechnology Information, Bethesda, MD, USA). Reactions were carried out on an Eppendorf RealPlex2 qPCR system, and fold changes in gene expression were calculated using the ΔΔCT method using species-specific GAPDH primers. Primer sequences are listed in Table S1. Human collagen 1 and α-SMA expression levels in the 2D FibCon condition were used as the reference group for fibroblast gene

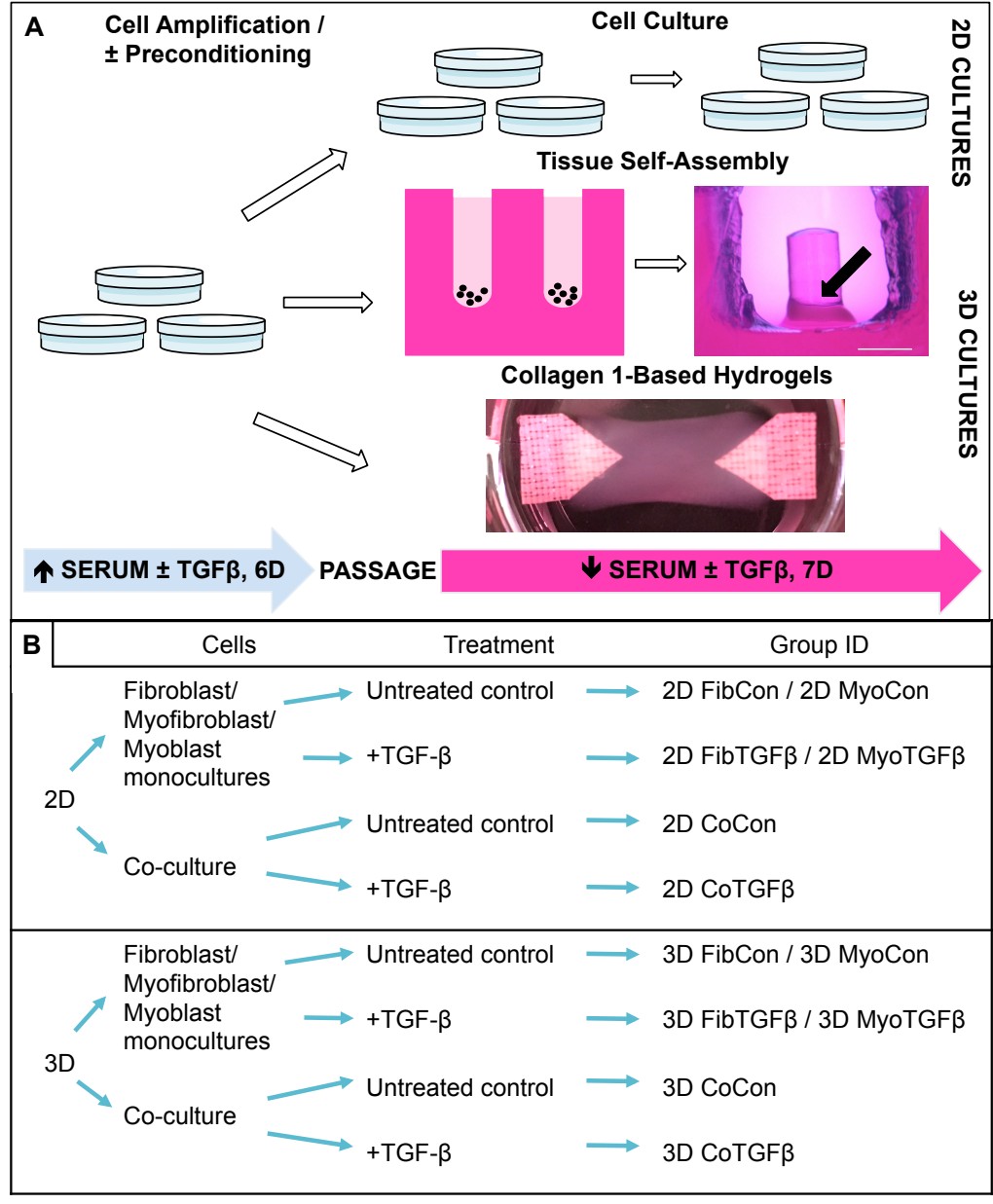

**Figure 1** **Technical protocol for 2D/3D culture systems and experimental conditions design.** (A) *Technique*: 2D cell culture and 3D tissue engineering protocols involve an initial high serum culture period on standard cell culture plates to separately amplify myoblast and fibroblast populations, with a subset of 'preconditioned' fibroblasts being treated with TGF-β1 to differentiate them into myofibroblasts. After 6 days the cells are passaged and either replated on cell culture plates for 2D studies or used for 3D systems. 3D tissue-engineered models were produced either via scaffoldless self-assembly or collagen 1 hydrogel biofabrication. A low serum culture period subsequently followed, with some groups treated with TGF-β1 over 7 days. The black arrow denotes the location of a tissue construct around the annulus inside an agarose mold; scale bar = 2 mm. (B) *Conditions*: experimental design is defined by comparisons between culture systems (2D culture plates vs 3D agarose gels/collagen 1-based hydrogels), cellular content (co-culture or monoculture of myoblasts, fibroblasts, or myofibroblasts), and their biochemical treatment in culture (±TGF-β1). The resulting group identification terms code for the conditions investigated in these experiments.

expression, and mouse myogenin gene expression levels of 2D MyoCon were used as a reference for myoblasts. $N = 5$ for all groups.

## Immunostaining

2D and 3D cultures were assessed qualitatively with immunostaining. 2D cultures were fixed with 4% formaldehyde for 20 min, and 3D cultures were fixed for 2 h. 3D self-assembled tissues were then submerged in OCT media, frozen for cryosectioning, sectioned at 30 μm thick, and applied to glass slides. Hydrogels remained unsectioned for whole-mount imaging. After fixing, 2D and 3D samples were permeabilized with 0.1% Triton X-100 in PBS for 20 min, and then incubated with a blocking solution consisting of 5% FBS in PBS for 20 min. Following blocking, samples containing myoblasts were stained with rhodamine phalloidin (R415, 1:40 dilution; Life Technologies, Carlsbad, CA, USA) and/or myosin heavy chain (MF-20, 1:50; Developmental Studies Hybridoma Bank, Iowa City, IA, USA), procollagen (M-38, 1:50; Developmental Studies Hybridoma Bank) or α-SMA (A 2547, 1:1,000; Sigma-Aldrich, St. Louis, MO, USA) primary antibodies. Primary antibody staining was followed by incubation with Alexa-Fluor 488 or 564 secondaries (1:250; Life Technologies, Carlsbad, CA, USA) and DAPI (1:50,000; Invitrogen, Carlsbad, CA, USA). Imaging of 2D samples was completed with an Olympus IX81 microscope (Olympus corporation of the Americas, Center Valley, PA, USA), 3D self-assembled tissues were imaged with an Olympus Fluoview FV1000 Confocal Microscope, and hydrogels were imaged with an Olympus Fluoview FV1000MPE multiphoton microscope. All images were processed with ImageJ (Version 1.48; National Institute of Health, Bethesda, MD, USA). The total area of procollagen staining was measured in 2D FibCon and FibTGFβ images and divided by the total number of nuclei to find the average area of procollagen staining per cell. Increased area procollagen signal was extrapolated as a change of in cell shape and increased cell spreading. Circularity of DAPI stained fibroblast nuclei was measured to compare changes in cyto- and nucleoskeletal shape. $N = 13$ images for each group. 2D MyoCon and CoCon images were also processed to measure changes in myotube alignment. In each image, the myotubes formed angles with a common axes ($x$ or $y$). The standard deviations of the angles from each image were measured to compare the variability in each group, with smaller averaged standard deviations for one sample image indicating higher myotube alignment. $N = 8$–$9$ images for each sample.

## Slack tests

The slack test method has been adapted by our lab to indirectly measure and compare tissue integrity, ECM maturity, and muscle fiber development between engineered samples. Two slack test methods were employed. In the whole tissue slack test, self-assembled tissue constructs were released from tension in culture and their shortening lengths were recorded over time. The initial lengths (μm, $L_i$) of rings were recorded across their longest axis while they remained on their posts. Rings were then removed from the agarose posts, suspended in PBS (37 °C) on a submerged horizontal microbeam from the Microsquisher system (CellScale, Ontario, Canada), and points across their longest axis for 27 min. The final lengths ($L_f$) were recorded and percent shortening (%) was calculated with the formula

% $= 100^*(L_f/L_i)$. Tissues were then stored in their respective medias for 24 h at 37 °C and imaged with a stereoscope. $N = 3$–6 samples per group.

The cleaved tissue slack test was performed to release the tissues from internal tension, which may arise from tissue structuring patterns building passive forces. 3D MyoCon, MyoTGFB, CoCon, and CoTGFB tissue rings were cut with a scalpel to generate linear muscle constructs freed from their agarose posts and placed in a 37 °C bath of cell culture media. Tissue lengths were measured every 3 min for 15 min, then samples were allowed to slack over 24 h at 37 °C to give the final length ($L_f$). The initial length (µm, $L_i$) for each construct, the length of the uncut tissue sample still mounted on its post, was calculated as the circumference of the 2 mm agarose posts using the formula: $L_i = 2\pi r$. The percent of original size (%) was determined with the formula: % $= 100^*(L_f/L_i)$. $N = 4$–5 samples per group.

## Tissue elastic moduli assessments

For calculations of the elastic stiffness of self-assembled engineered tissues, two methods of mechanical characterization were used: a compression-based system to generate slopes from stress–strain curves and atomic force microscopy (AFM). Tissue constructs were sectioned into 4–5 pieces with diameters between 250 and 500 µm and were compressed in a PBS fluid bath (pH 7.4, 37 °C) using a Microsquisher (Cellscale, Waterloo, Ontario, Canada). Tissues were compressed to 40% of their original diameter at a rate of 1% per second using microbeams with diameters between 0.2 and 0.3048 mm. Force–displacement curves were generated and the slope of the linear portion of the curve was extrapolated as the elastic modulus of the tissue. Slopes were averaged for each 3D condition with $N = 8$–12 section samples per group.

Tissue elastic moduli were measured by an Asylum MFP3D-Bio AFM (Asylum Research, Santa Barbara, CA, USA) through a nano-indention method using MFP-3D software (Version 13.04.77). Force-distance curves were determined using Eqs. (1) and (2):

$$F(\delta) = \frac{4\sqrt{R}}{3} \frac{E}{1-\nu^2} \delta^{3/2} \tag{1}$$

$$F(\delta) = \frac{E}{1-\nu^2} \frac{2\tan\alpha}{\pi} \delta^2. \tag{2}$$

Where $F$ is the measured force, $E$ is the local Young's modulus, $R$ is the cantilever's tip radius (for a spherical tip), $\alpha$ is the cantilever's tip angle (for a cone tip), $\nu$ is the Poisson's ratio of the sample (assumed as 0.5 for an incompressible material), and $\delta$ is the sample indentation. Pyramidal tips with a nominal tip radius 20 nm, 200 µm in length, 20 µm in width, and a tip semi-angle of 15° on silicon nitride triangular V-shaped cantilevers with a nominal spring constant of 0.06 N/m (DNP-10; Bruker Inc., Camarillo, CA, USA) were employed. The recorded force-distance curves were analyzed in MATLAB and statistical analysis was done using SPSS (Ver. 17.0; IBM, Somers, NY, USA).

## Statistics

Statistical analyses were performed with two-tailed unpaired *t*-tests and one-way ANOVAs using GraphPad Prism (Version 4; GraphPad Software) and statistical significance was defined as $p < 0.05$. Means are reported with standard error bars in bar graphs. Skew coefficients were calculated in Excel (Version 14.6.4; Microsoft).

## RESULTS

### Fibroblast to myofibroblast differentiation is observed in 2D cultures treated with TGF-β1, regardless of myoblast presence

2D fibroblast-only cultures displayed increased α-SMA staining in TGF-β1-treated samples (Figs. 2A and 2B) and increased α-SMA gene expression ($p < 0.01$, Fig. 3A), indicating their differentiation to myofibroblasts. There was no significant difference in collagen 1 transcription between any 2D conditions containing fibroblasts or myofibroblasts (Fig. 3B), but post-translational control of the protein differed with TGF-β1 treatment. Fibroblasts in the control group were spindle-shaped and displayed smaller areas of procollagen signal per cell nuclei (Figs. 2C and 2I), while TGF-β1 treatment significantly increased the area of procollagen staining per nuclei by 32.2 % ($p < 0.0001$, Figs. 2D and 2I) and enhanced the secretion of procollagen out of the cell (red arrow in 2D). Nuclei circularity was also increased in 2D FibTGFβ samples ($p < 0.0001$, Fig. 2J), indicating a widening of the nucleoskeleton accompanying increased cell spreading, compared to the more elongated nuclear shape in control conditions. Enhanced cell proliferation was also observed with TGF-β1 treatment ($p < 0.0001$, Fig. 2K), as recorded by average number of nuclei per field of view. α-SMA gene expression in 2D CoCon samples was similar to transcript levels of the fibroblast-only control group, but was significantly upregulated in 2D CoTGFβ conditions ($p < 0.01$, Fig. 3A).

### Myogenesis is suppressed in 2D cultures with TGF-β1 supplementation and in the presence of fibroblasts and myofibroblasts, yet fibroblasts increase myotube alignment

Robust myotube formation was present in 2D MyoCon samples (Fig. 2E), decreased with exposure to TGF-β1 (Fig. 2F) or in co-culture with fibroblasts (Fig. 2G), and completely inhibited in myofibroblast co-cultures supplemented with TGF-β1 (Fig. 2H). MYOG gene expression in the 2D MyoTGFβ and 2D CoCon conditions were similar to each other and both displayed significant MYOG downregulation in comparison to 2D MyoCon. 2D co-cultures exposed to TGF-β1 showed the lowest MYOG expression of any group ($p < 0.01$, Fig. 3C). Despite diminishing myogenesis, fibroblasts organized myotube formation by increasing their alignment in 2D CoCon samples ($p < 0.001$, Fig. 2L).

### In comparison to fibroblasts grown on plastic, self-assembled tissues containing only fibroblasts have a suppressed ability to assume a myofibroblast phenotype

In comparison to 2D fibroblast-only controls, expression of procollagen was dramatically downregulated in fibroblast-only tissues, and this was not improved by TGF-β1 exposure

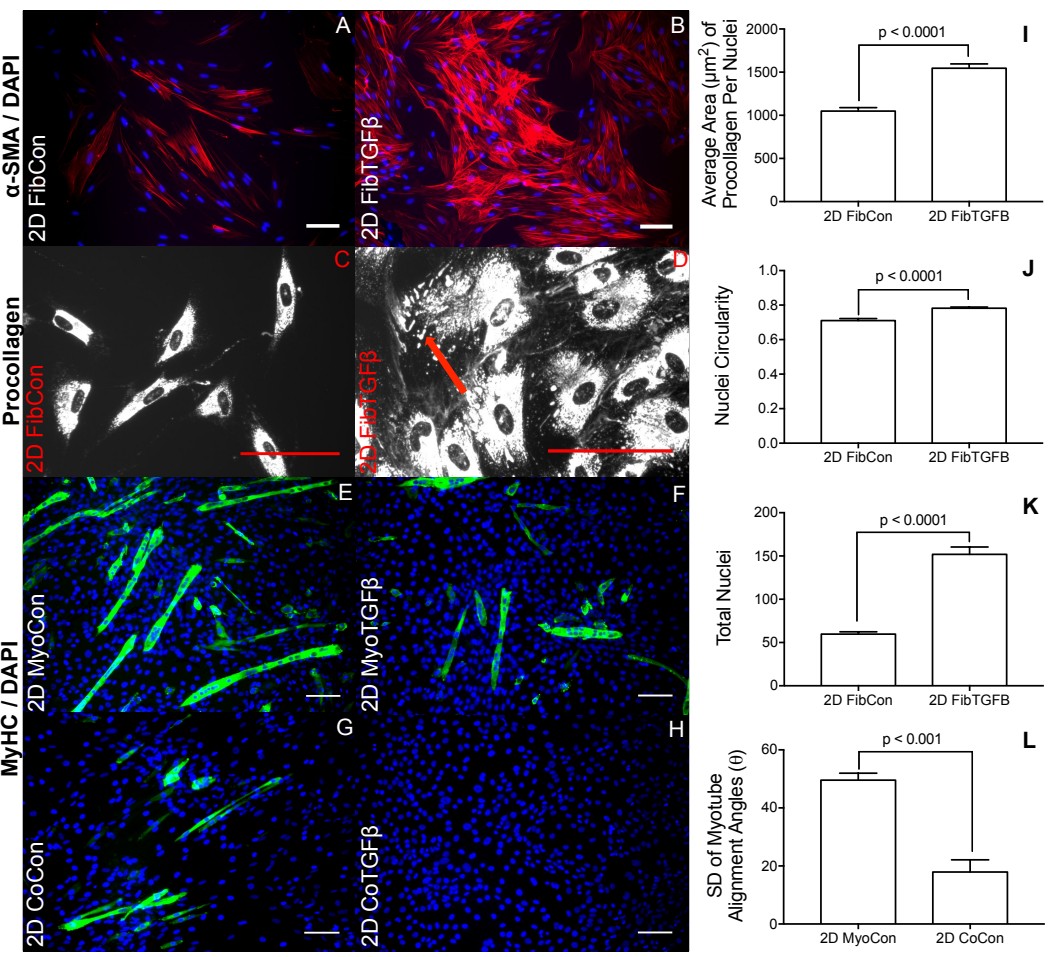

**Figure 2** **Immunostaining and morphological characterization of 2D cultures.** 2D FibCon (A) and FibTGFβ (B) stained for α-SMA and DAPI, and FibCon (C) and FibTGFβ (D) stained for procollagen. 2D MyoCon (E), MyoTGFβ (F), CoCon (G), and CoTGFβ (H) conditions stained for myosin heavy chain (MyHC) and DAPI. Scale bars = 100 µm for all images. Quantitative analysis of average area of procollagen signal/average number of nuclei per field of view (I), nuclei circularity (J), and average number of total nuclei per field of view (K) in FibCon and FibTGFβ samples. Quantitative analysis of myotube alignment (L) in MyoCon and CoCon cultures, reported as the standard deviation of myotube alignment angles. Graphs display group averages with standard error bars.

($p < 0.01$, Fig. 3B). α-SMA staining was absent in 3D FibCon samples (Fig. S1A), but was somewhat increased with TGF-β1 treatment (Fig. S1C). 3D FibCon and FibTGFβ α-SMA gene expression was transcriptionally downregulated to an even greater degree than procollagen, yet these values were not significantly different from each other ($p < 0.01$, Fig. 3A). Fibroblast/myofibroblast-only tissues were quite fragile; they would easily rupture during handling (arrow, Fig. 4D) and would sometimes collapse, losing their annular shape (Figs. 4C and 4F). Despite their fragility, their surfaces were smooth and homogenous (Fig. 4S). However, the average thickness of constructs containing myofibroblasts treated with TGF-β1 was 33.2% larger than respective controls ($p < 0.01$, Fig. 5F), similar to the myofibroblast hypertrophy seen in 2D FibTGFβ.

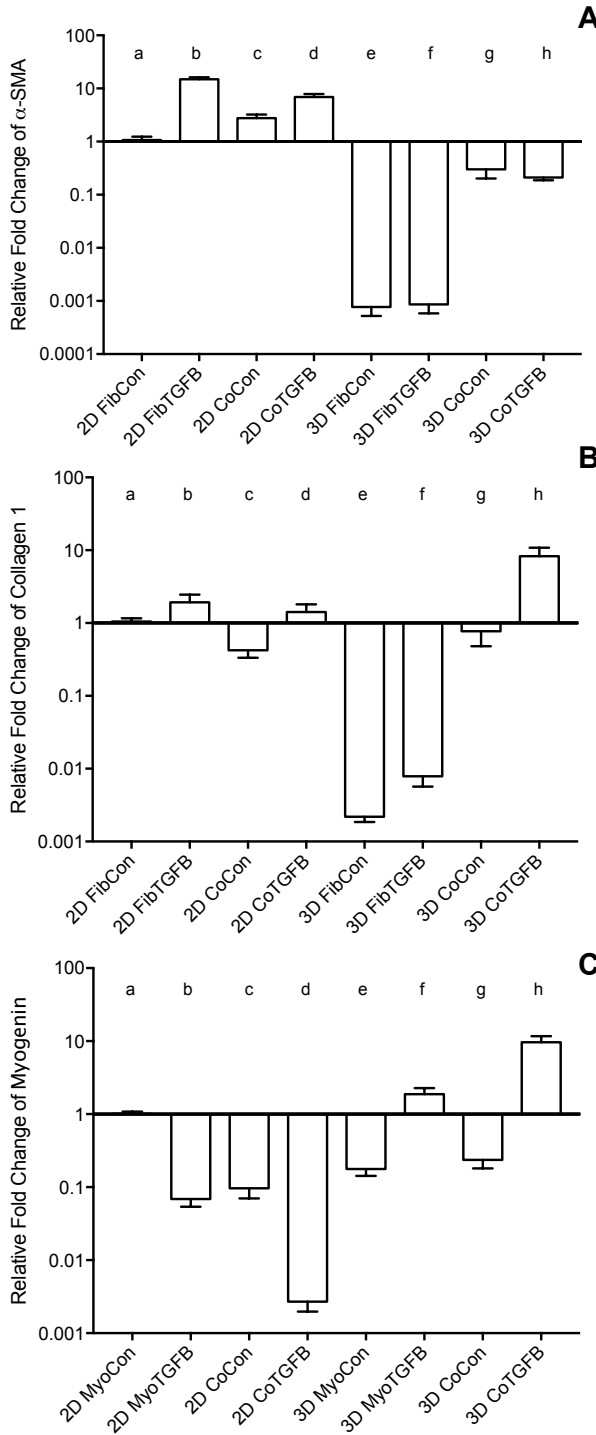

**Figure 3** **Gene expression of fibroblast collagen 1 and α-SMA and myoblast myogenin in 2D cell cultures and 3D self-assembled tissue conditions.** Species-specific gene expression for all groups is normalized to gene expression for group (a), 2D FibCon or MyoCon samples. (continued on next page...)

**PeerJ** ________________________________________________________

**Figure 3 (…continued)**
Average fold change values are reported with standard error bars. Significance values were defined at $p <$ 0.01. (A) α-SMA: (a) is significantly different from (b), (d), (e), (f), (g), and (h); (b) is significantly different from (c), (e), (f), (g), and (h); (c) is significantly different from (e), (f), (g), and (h); (d) is significantly different from (e), (f), (g), and (h); (e) is significantly different from (g) and (h); and (f) is significantly different from (g) and (h). (B) COL1: (a) is significantly different from (e), (f), and (h); (b) is significantly different from (c), (e), (f), and (h); (c) is significantly different from (e), (f), and (h); (d) is significantly different from (e), (f), and (h); (e) is significantly different from (g) and (h); (f) is significantly different from (g) and (h); and (g) is significantly different from (h). (C) MYOG: (a) is significantly different from (b), (c), (d), (e), (g), and (h); (b) is significantly different from (d), (f), (g), and (h); (c) is significantly different from (d), (f), and (h); (d) is significantly different from (e), (f), (g), and (h); (e) is significantly different from (f) and (h); (f) is significantly different from (g) and (h); and (g) is significantly different from (h).

## TGF-β1 supplementation improves myoblast differentiation and alignment in myoblast-only self-assembled tissues

3D myoblast-only groups differed in their degree of myogenesis and their myotube formation patterns. 3D MyoCon contained myotubes that had some degree of organization but generally weren't aligned (Figs. 6A and 6B), whereas myoblast samples treated with TGF-β1 had myotubes that were aligned along their circumferential axis (white filled arrow, Figs. 6E and 6F). Gene expression profiles indicated that 3D MyoCon samples had significantly lower myogenin expression than both the 2D MyoCon control and 3D MyoTGFβ condition ($p < 0.01$, Fig. 3C), but the 2D control and 3D TGFβ treated condition were not significantly different from each other. α-SMA was present in both untreated (Figs. 6C and 6D) and TGF-β1 treated (Figs. 6G and 6H) myoblast constructs. Observable in the α-SMA stained sections are fissures and breaks in the tissue (white asterisks, Figs. 6C and 6G). Additionally, the surfaces of myoblast-only tissues presented nodular syncytium of fused myoblasts lacking the anchorage necessary to elongate and form myotubes (arrow, Figs. 6I–6K), and this was macroscopically observable (unfilled arrows, Fig. 4T). Heterogeneous tissue patterning is also seen here, with many ripples along the tissue surface, large budges (filled arrow, Fig. 4T), and regions of varying density.

## In self-assembled co-cultures, fibroblasts and myofibroblasts homogenize tissue surfaces, myofibroblasts improve myotube formation, and TGF-β1 enhances myogenesis

Addition of fibroblasts or myofibroblasts smoothened the surface of co-culture tissues (Fig. 4U), similarly to fibroblast- or myofibroblast-only samples, and these constructs lacked the studding of nodular syncytium seen in myoblast-only tissues. Co-culture conditions (Figs. 4M and 4P) appeared visually denser than their fibroblast- (Figs. 4A and 4D) and myoblast-only (Figs. 4G and 4J) counterparts. Similar to the hypertrophy observed in 2D and 3D myofibroblast conditions treated with TGF-β1, CoTGFβ tissues were 27.9% thicker than CoCon ($p < 0.01$, Fig. 5F). While human fibroblast/myofibroblast α-SMA gene expression was decreased in all 3D conditions compared to the 2D FibCon group, it was significantly higher in 3D CoCon and CoTGFβ compared to 3D fibroblast-only tissues ($p < 0.01$, Fig. 3A), and co-culture transcription levels were not influenced by TGF-β1 supplementation. A non-species specific α-SMA antibody was visible within

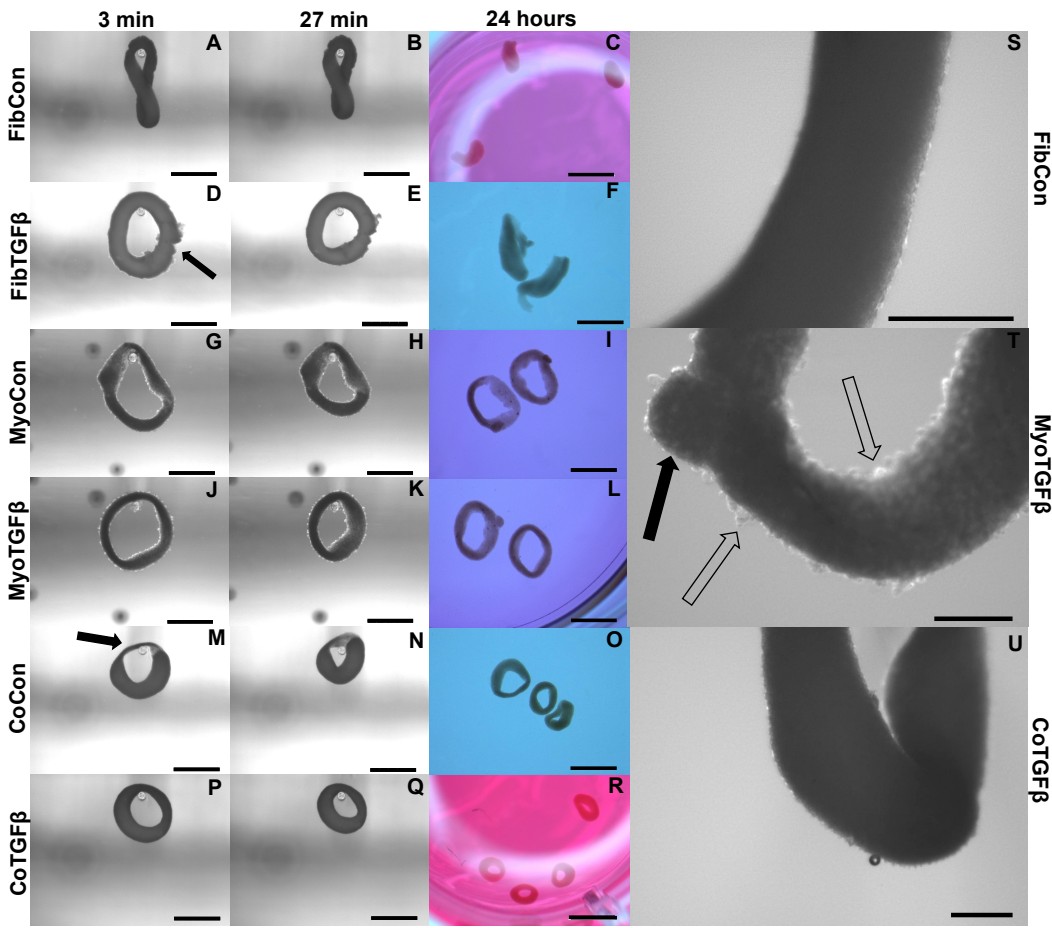

**Figure 4** **Zero force shortening of self-assembled tissues and their surface characteristics.** Tissue shortening was assessed when removed from agarose molds after 3 min, 27 min, and 24 h. 3D FibCon (A–C), FibTGFβ (D–F), MyoCon (G–I), MyoTGFβ (J–L), CoCon (M–O), and CoTGFβ tissue rings (P–R), with scale bars = 1,000 μm. The filled arrow in (D) indicates a structural rupture in the tissue, and the black arrow in (M) indicates thinned myoblast-free region of sample. Higher magnification images of FibCon (S), MyoTGFβ (T), and CoTGFβ tissues (U) show surface texture. In (T), the filled arrow is a large nodule on a MyoTGFβ sample, and unfilled arrows indicate smaller syncytium; scale bars = 300 μm.

myotubes, where CoCon samples showed α-SMA within punctate and short myotubes (Fig. 7A), while those in CoTGFβ tissues were elongated and more mature (Fig. 7C), indicating that TGF-β1 facilitated the development of bundles of myotubes aligned in parallel. Additionally, collagen 1 expression was increased in 3D co-cultures compared to 3D fibroblast/myofibroblast-only cultures ($p < 0.01$, Fig. 3B), but CoCon constructs were not significantly different from the 2D FibCon condition. Interestingly, collagen 1 gene expression was highest in 3D CoTGFβ samples, which was also the condition of highest MYOG expression ($p < 0.01$).

3D CoCon constructs containing fibroblasts frequently presented dense bulges with thickened regions of cells composed of differentiating myoblasts (Figs. 7E and 7F) between thinner regions absent of MyHC staining (Figs. 7H and 7I); this heterogeneous tissue
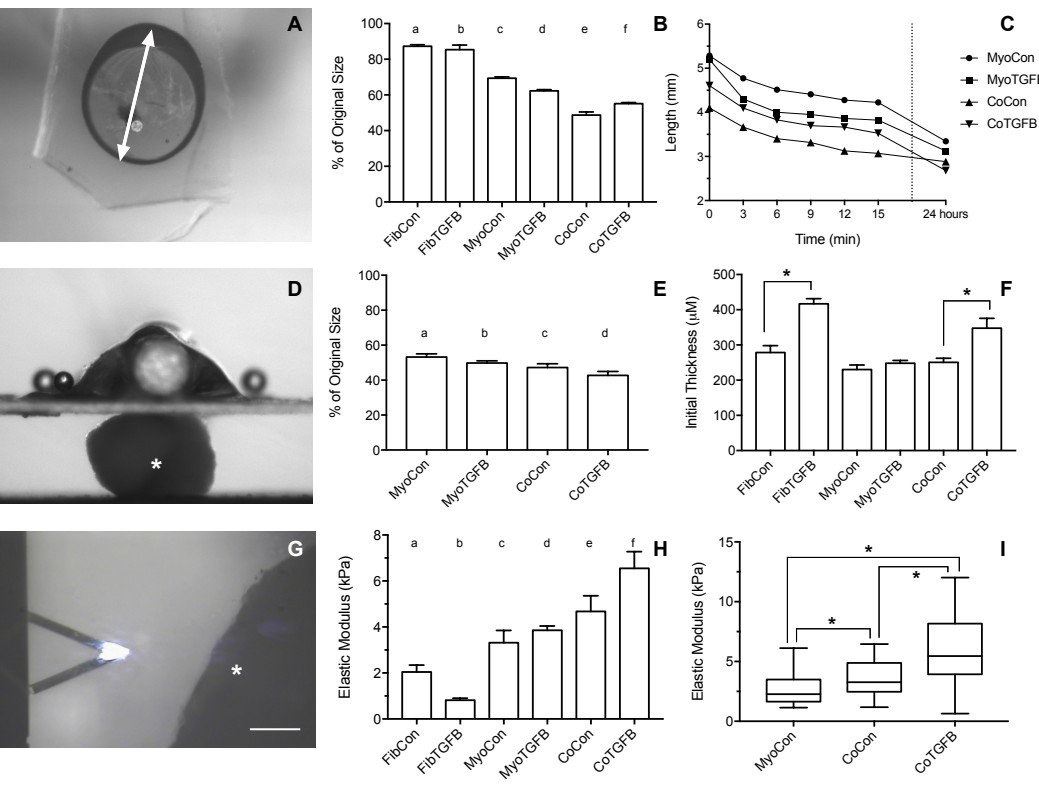

**Figure 5** **Biomechanical and elastic properties of 3D tissue constructs.** (A) An agarose gel mold with central post and tissue of tissue skewered onto a microbeam from the Microsquisher system. The white arrow denotes the longest axis of the tissue constructs used to track slacking from initial length in the whole tissue slack test, scale bar = 1,000 µm. (B) Percent (%) of initial length of the longest axis of tissue samples in the whole tissue slack test 27 min after their removal from agarose gels. All groups shortened from their original size significantly ($p < 0.001$), with significant differences between each group ($p < 0.05$), except for (a) and (b). (C) The shortening length of tissue samples over 3 min intervals in the cleaved tissue slack test. Note the final data point is at 24 hours, and the $Y$-axis origin is 2 mm. (D) Image of the Microsquisher system used to obtain elastic's modulus of tissue sections (white asterisk); scale bar = 300 µm. (E) Percent (%) of initial length of tissue samples in the cleaved tissue slack test after 24 h. All groups shortened from their original size significantly ($p < 0.001$), with significant differences between each group ($p < 0.05$), except for (b) and (c). (F) Initial thickness of tissue samples, asterisks indicate $p < 0.001$. (G) Screenshot of AFM cantilever and sample (white asterisk), scale bar = 100 µm. (H) Young's modulus of tissues generated from Microsquisher force-displacement data. (a) is significantly softer than (e) and (f); (b) is significantly softer than (c), (d), (e), and (f); and (c) and (d) are significantly softer than (f), with $p < 0.001$. (I) Young's modulus of MyoCon, CoCon, and CoTGFβ tissues calculated by AFM, asterisks indicate $p < 0.01$. Graphs display group averages with standard error bars.

architecture can also be seen macroscopically (arrow, Fig. 4M). CoCon constructs contained myotubes that were short but numerous. Some myotubes fused from one wall of the tissue to another, so that during sectioning, cross sections of myotubes are apparent (filled arrows, Figs. 7E and 7F). In contrast, CoTGFβ samples containing myofibroblasts had myotubes that were thicker and more multinucleated (Figs. 7G, 7J and 7K). These tissues also had many cross-sections of myotubes (filled arrow, Fig. 7J), but additionally contained much longer circumferentially aligned myotubes than in control co-culture samples (unfilled

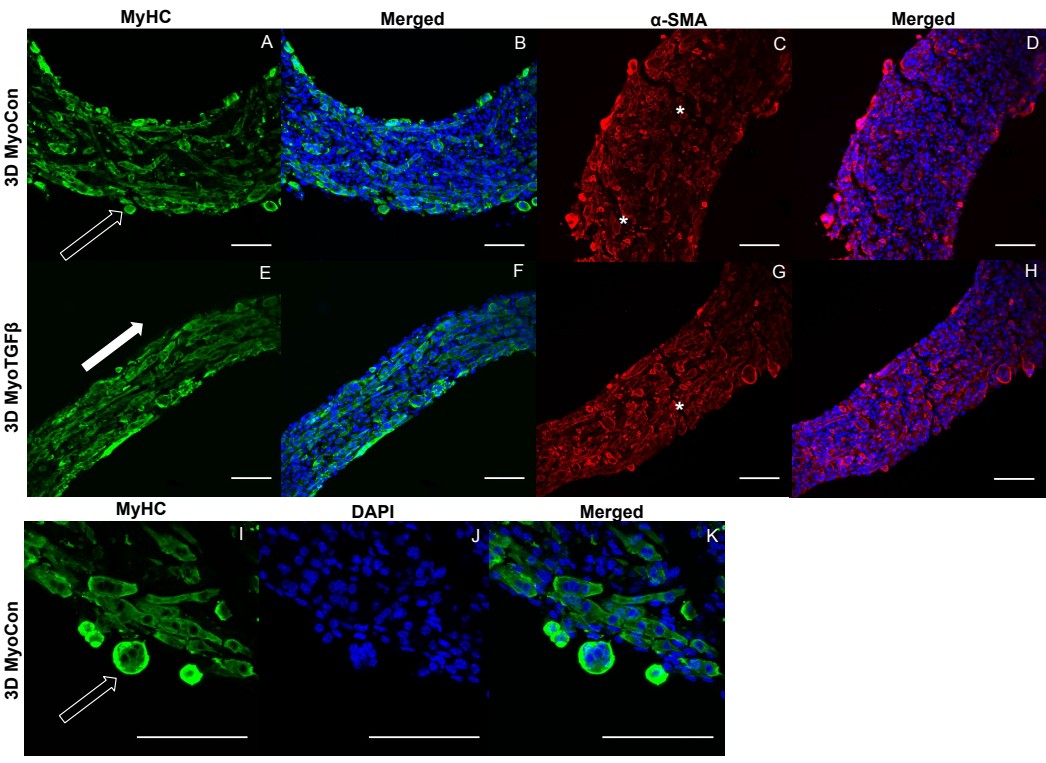

**Figure 6  Histology of self-assembled myoblast monocultures.** 3D MyoCon (A–D, I–K) and MyoTGFβ constructs (E-H) are displayed with myosin heavy chain (A, E, I), MyHC merged with DAPI (B, F, K), and DAPI alone (J). α-SMA (C, G) and α-SMA merged with DAPI (D, H). White filled arrows indicate direction of alignment and anisotropy of myotubes (E), while unfilled arrows indicate unanchored syncytium (A, I), and white asterisks identify regions of tissue breakage (C, G), scale bars = 100 μm.

arrow, Fig. 7G). MYOG expression in 3D CoTGFβ constructs was significantly higher than all other groups ($p < 0.01$, Fig. 3C), while MYOG expression in 3D CoCon constructs was suppressed compared to 2D MyoCon ($p < 0.01$), and was not significantly different from 3D MyoCon. This data demonstrates that addition of fibroblasts with myoblasts in 3D co-culture does not improve myogenesis beyond what is observed in cultures consisting solely of myoblasts, and TGF-β1 supplementation is a myogenesis-promoting factor.

## Zero-force velocity is greatest in co-culture with TGF-β1 supplementation in slack tests

Slack tests are a measurement of zero force velocity tissue shortening resulting from the recoil of stretched elastic elements and myofibrillar filament sliding in sarcomeres. Slack tests assess acto-myosin kinetics that are independent of $Ca^{2+}$ activation, which provides biomechanical information about extracellular matrix networking, MHC isoform and muscle fiber type, and muscle organ characteristics (*Claflin & Faulkner, 1985*; *Josephson & Edman, 1998*; *Reggiani, 2007*). In whole tissue slack tests, the initial maximum lengths of self-assembled tissue constructs were recorded prior to removal from their posts (white arrow, Fig. 5A). Constructs were then released from tension (Figs. 4A, 4D, 4G, 4J, 4M

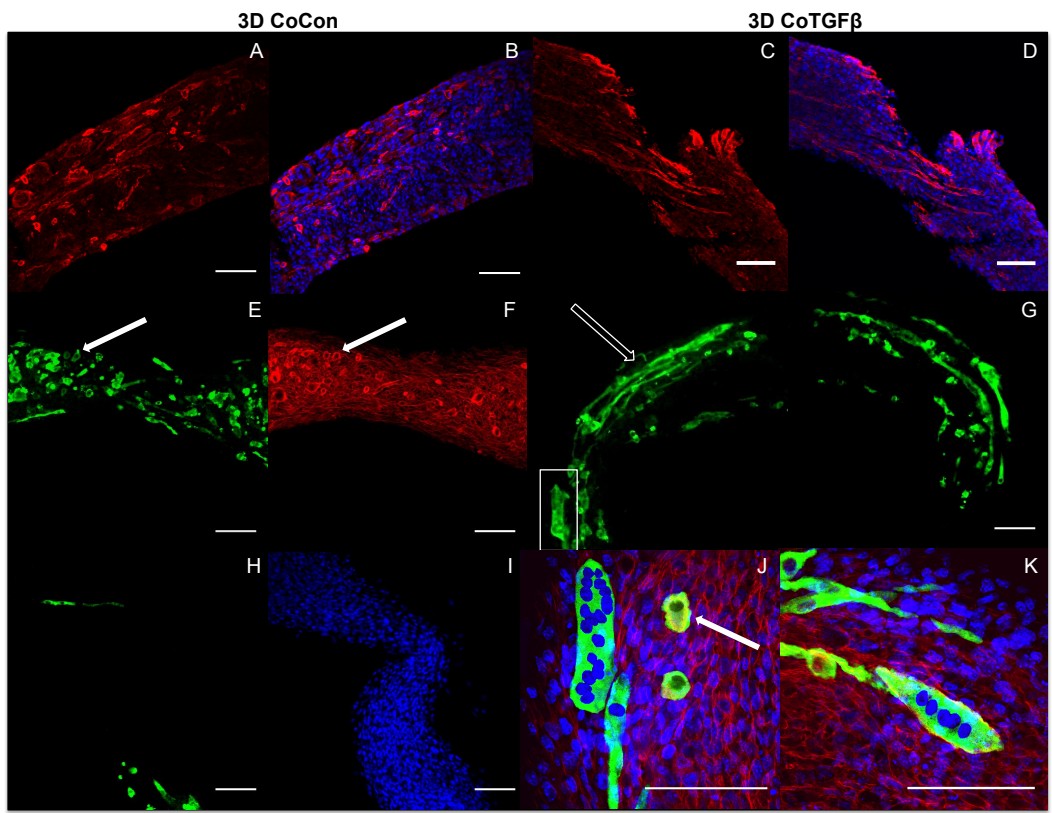

**Figure 7** **Histology of self-assembled co-cultures.** 3D CoCon (A, B, E, F, H, I) and CoTGFβ tissues (C, D, G, J, K). α-SMA (A, C) and α-SMA merged with DAPI (B, D). Myosin heavy chain staining of CoCon constructs (E, H), f-actin staining of E (F), and DAPI staining of H (I). MyHC staining of CoTGFβ tissues (G, J, K) with DAPI and f-actin staining (J, K). (J) is an inset of G. White arrows indicate cross sections of myotubes observed with MyHC and actin staining, while unfilled arrows show elongated myotubes fused along circumferential axis of tissue rings, scale bar = 100 μm.

and 4P), their lengths measured after 27 min (Figs. 4B, 4E, 4H, 4K, 4N and 4Q), and allowed to slack over 24 h (Figs. 4C, 4F, 4I, 4L, 4O and 4R). While all groups shortened significantly from their original size after 27 min ($p < 0.001$, Fig. 5B), tissues containing only fibroblasts or myofibroblasts shortened the least, with 3D FibCon remaining 87.3% and FibTGFβ 85.4% their original size. Fibroblast/myofibroblast tissues were the only cell-content matched tissue group that did not display significantly different final length values due to TGF-β 1 treatment. Myoblast tissues underwent significantly greater shortening than fibroblast/myofibroblast samples: MyoCon rings shrank to 69.4% and MyoTGFβ to 62.2% their initial diameter. Co-culture tissues underwent the greatest alteration, however. CoCon samples had the most significant shortening compared to all other groups (48.7%), and CoTGFβ was the second most shortened group (55.1%). Interestingly, after 24 h, CoTGFβ tissues (Fig. 4R) showed a greater degree of sample shortening than CoCon (Fig. 4O).

Since reference points for tracking changes in length were not conserved overnight in ring-shaped tissues, a cleaved tissue slack test was performed by cutting tissues containing

C2C12s into linear constructs and plotting slacking lengths over time (Fig. 5C). Most of the length shortening occurred within the first 3 min and slowed considerably by 9 min, with co-culture samples displaying the highest velocities. After 24 h, 3D MyoCon samples shortened to 53.2% of their initial length, MyoTGFB to 49.8%, CoCon to 47.2%, and CoTGFB to 42.7% (Fig. 5E). Each experimental group was significantly different from the others ($p < 0.01$), except for MyoTGFB and CoCon. The slack tests demonstrated that while TGF-β1 exposure for fibroblast-only samples did not lead to significantly more zero-force shortening in comparison to untreated controls, TGF-β1 positively impacted shortening of myotubes within myoblast-containing tissues, and the interactions between TGF-β1-treated myofibroblasts and myoblasts promoted the greatest degree of shortening over time.

## Tissue elastic moduli increase in co-cultures and with TGF-β1 exposure

Using the parallel plate compression system, the Young's modulus for sectioned self-assembled tissues was determined to be 2.04 kPa for FibCon, 0.81 kPa for FibTGFβ, 3.32 kPa for MyoCon, 3.86 kPa for MyoTGFβ, 4.47 kPa for CoCon, and 6.55 kPa for CoTGFβ (Fig. 5H). FibCon and FibTGFβ tissues were significantly softer than CoCon and CoTGFβ ($p < 0.001$). While MyoCon and MyoTGFβ were also significantly softer than CoTGFβ ($p < 0.001$), TGF-β1 did not significantly increase stiffness of samples containing only myoblasts. Similarly, while TGF-β1 treatment increased the stiffness of CoTGFβ, it was not a significant increase with respect to CoCon. Consequently, AFM was used to increase the sensitivity of measurements of elasticity at the tissue surface. AFM generated Young's moduli of 2.66 kPa for MyoCon, 3.64 kPa for CoCon, and 6.02 kPa for CoTGFβ, with all groups being significantly different from one another ($p < 0.01$, Fig. 5I).

## Supplementation with TGF-β1 and addition of myofibroblasts enhances myogenesis in collagen 1-based hydrogels

MyoCon, MyoTGFβ, CoCon, and CoTGFβ hydrogels were imaged for myosin heavy chain. In the myoblast control condition lacking TGF-β1 supplementation, some MyHC was present in unfused mononuclear cells, but observable myotube formation was absent (Fig. 8A). In contrast, TGF-β1-treated myoblast hydrogels showed robust myotube formation, with somewhat inconsistent alignment orientations (Fig. 8B). Without supplementation with TGF-β1, co-cultures with fibroblasts and myoblasts showed small and immature myotube-shaped cells (Fig. 8C), in contrast to its TGF-β1-treated counterpart containing myofibroblasts (Fig. 8D). CoTGFβ hydrogels yielded the thickest and longest muscle fibers of any condition and possessed the highest degree of multinucleation. These muscle fibers also demonstrated the best alignment. CoTGFβ hydrogels were also more visibly contracted in their culture wells than other conditions (Fig. 8E), although MyoTGFβ hydrogels displayed a lesser degree of contraction. Contraction of MyoCon and CoCon hydrogels was not observed.

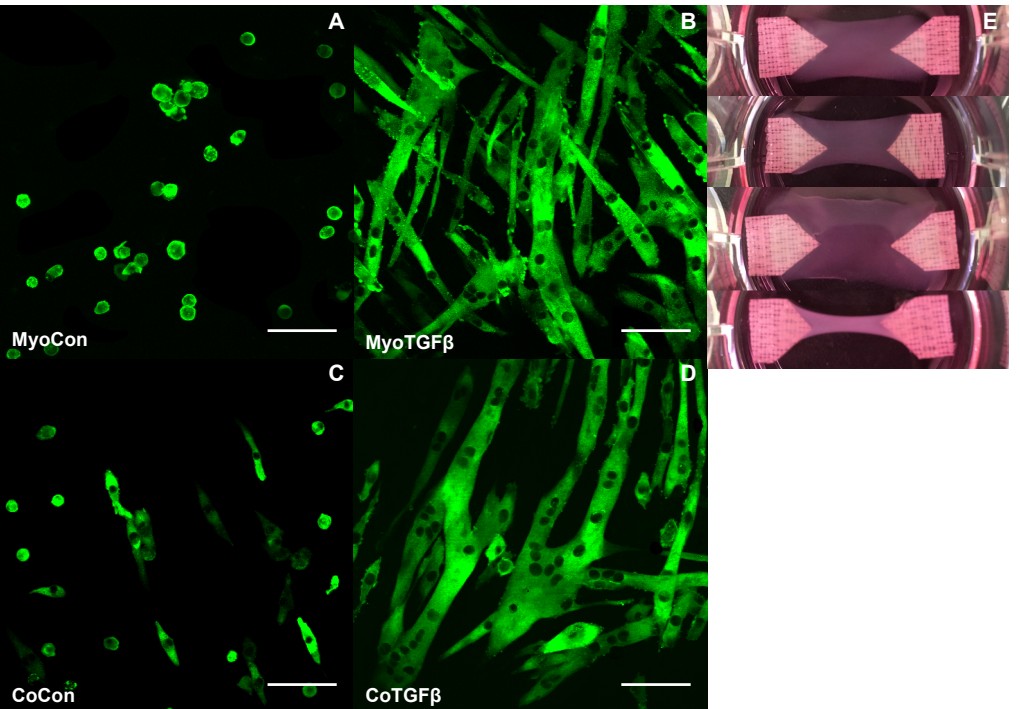

**Figure 8** **Histology of mono- and co-cultures in collagen 1-based hydrogels.** Myosin heavy chain staining of MyoCon (A), MyoTGFβ (B), CoCon (C), and CoTGFβ (D) hydrogels. Scale bar = 100 μm. (E) are images of the hydrogels from culture wells of a 6-well plate, in descending order: MyoCon, MyoTGFβ, CoCon, and CoTGFβ hydrogels.

## DISCUSSION

These experiments investigated the impact of tissue remodeling and ECM deposition on the biomechanical tissue niche; TGF-β1 signaling; and cell–cell signaling between fibroblasts, myofibroblasts, and myoblasts on *in vitro* simulations of myogenesis. Our data demonstrates that transitioning from 2D to 3D culture systems has a profound impact on gene and protein expression involved in mechanotransduction. Consistent with other literature, TGF-β1 differentiated fibroblasts to myofibroblasts in 2D cultures containing only fibroblasts (*Dahl, Ribeiro & Lammerding, 2008*; *Dong et al., 2013*; *Sandbo & Dulin, 2011*). This was accompanied by an upregulation of α-SMA gene and protein expression. While myofibroblasts in 2D did not show upregulated transcription levels of collagen 1, post-transcriptional control mechanisms increased collagen translation and secretion into the extracellular space. Steady collagen gene expression levels in 2D may be maintained by mechanotransduction signaling, which was altered in the equivalent 3D self-assembled tissue model, and resulted in radical suppression of collagen 1 and α-SMA transcription.

Additionally, myoblasts have been reported to assume a myofibroblast-like phenotype with TGF-β1 exposure in 2D (*Cencetti et al., 2010*; *Charge & Rudnicki, 2004*; *Filvaroff, Ebner & Derynck, 1994*; *Li et al., 2004*), and our TGF-β1 treated 2D cultures showed similar decreases in myogenecity in monocultures, with the myofibroblast co-culture displaying

the most downregulated myogenin expression of any group. *In vivo,* however, while muscle fiber regeneration can be inhibited by exogenous TGF-β1 (*Filvaroff, Ebner & Derynck, 1994*; *Mendias et al., 2012*), endogenous TGF-β1 signaling plays an important role in myogenesis. Genetically truncating the type II TGF-β receptor in myoblasts arrests their ability to differentiate *in vivo*, indicating that TGF-β1 signaling contributes to muscle development (*Karalaki et al., 2009*; *Myhre & Pilgrim, 2012*). In agreement with these studies, we found that TGF-β1 supplementation improved myogenesis in our 3D mono- and co-culture muscle models, a reversal of the observations in 2D cultures.

The mechanism for TGF-β1-mediated increases in myogenic differentiation and contractility highlights a differential effect of TGF-β1 on myofibrillogenesis in 2D and 3D systems. Myofibrillogenesis initiates sarcomere development by increasing the dense packing of contractile and cytoskeletal proteins in muscle cells. During myofibrillogenesis, myofibrils are assembled from the framework of the actin cytoskeleton and undergo maturation by successive incorporation of proteins with increasing contractility. This process is driven by transmitting mechanical tension in the actin cytoskeleton to surrounding ECM through costameres (*Sanger et al., 2010*) and results in the development and alignment of sarcomeres (*Weist et al., 2013*). TGF-β receptor activation increases cytoskeletal polymerization of α-SMA stress fibers in 2D culture platforms (*Filvaroff, Ebner & Derynck, 1994*), but also simultaneously has an inhibiting effect on myogenesis. However, since α-SMA is a precursor protein to sarcomeric actin in myofibrillogenesis, moderate TGF-β1 levels in 3D may accelerate myofibrillogenesis through an α-SMA-mediated upregulation mechanism that could increase the tensional forces cells exert on their environment.

In 3D, fibroblasts and myofibroblasts benefit muscle regeneration through depositing ECM, organizing myoblast differentiation and myotube fusion, and stimulating myofibrillogenesis (*Sanger et al., 2010*; *Turrina, Martinez-Gonzalez & Stecco, 2013*). Myotube formation dynamically remodels the biomechanical environment within muscle to further attenuate fibroblast and myofibroblast activity. *In vitro* cell traction studies have shown that myotubes exert five to eight times greater traction forces on their substrates than fibroblasts, and these forces increase as myotubes mature (*Dahl, Ribeiro & Lammerding, 2008*). Contracting myotubes transmit their forces to surrounding ECM through costameres and exert tensional forces throughout the muscle organ to facilitate contraction (*Costa, 2014*). This mechanism of force transmission could explain the greater recoil and shortening capacity co-culture tissues in contrast to myoblast-only samples with a suppressed ability to disperse forces to the surrounding ECM. Lower levels of supportive ECM for myotubes in self-assembled monocultures decreases the availability for integrin connections to the tissue microenvironment and reduces tensional forces applied to surrounding structures, resulting in less elastic recoil after being released from tension. Additionally, our data showing lower elastic moduli in 3D myoblast monocultures than co-cultures is congruent with the findings of Meyer and Lieber, who found the elastic stiffness of skeletal muscle fibers to be significantly dependent on the presence of ECM. In

their study, dissected muscle fibers stripped of their extracellular matrix sheaths had four-fold lower elastic moduli than fascicles that retained their ECM (*Meyer & Lieber, 2011*).

3D fibroblast and myofibroblast co-cultures were distinct in their morphology, force transmission capacities, and biomechanical properties. The bundling of highly multinucleated myotubes in parallel within CoTGFβ constructs was more similar to *in vivo* muscle fascicle morphology than fibroblast co-cultures, and was accompanied by the greatest upregulation in myogenin expression of all groups. Enhanced muscle fiber maturation and patterning likely increased the magnitude of zero-force shortening in CoTGFβ samples due to acto-myosin kinetics. This compliments a study by Larkin et al., who found that TGF-β1 treatment of heterogeneous fibroblast and satellite cell populations in engineered muscle constructs increased their force generation capacities during electrical stimulation (*Weist et al., 2013*). However, this study did not investigate the differentiation of fibroblasts to myofibroblasts within a co-culture system (*Weist et al., 2013*). Our experiments identify that TGF-β1 has both individual and synergistic benefits to skeletal muscle engineering. These data show that TGF-β1 treatment in 3D monocultures of myoblasts boosts myoblast fusion and differentiation. In co-cultures with myoblasts, TGF-β1 activates fibroblasts to differentiate into myofibroblasts, which are better accelerators of myogenesis than fibroblasts, due in part to their enhanced ECM synthesis. The tenacity of myofibroblasts in co-culture is demonstrated through the highest collagen 1 gene expression of all conditions, where collagen 1 reinforcement likely improved tissue integrity and stiffness. Here we have demonstrated a robust synergistic effect between the incorporation of myofibroblasts, not fibroblasts, and TGF-β1 supplementation in tissue-engineered models of skeletal muscle myogenesis.

## CONCLUSIONS

These experiments have demonstrated that fibroblasts and myofibroblasts are not interchangeable cell types in *in vitro* models. To the author's knowledge, we are the first to report that myofibroblasts enhance the outcome of myogenesis to a greater degree than fibroblasts when in co-culture with myogenic cells in a tissue engineered model of skeletal muscle. Additionally, our comparison between 2D and 3D studies highlights the importance of using a culture model that appropriately recapitulates the *in vivo* environment. In literature, many mechanistic, functional, and differentiation studies of myogenic cells treated with TGF-β1 performed in 2D lead to erroneous conclusions about the impact of TGF-β1 on myogenesis. We are the first to methodologically describe this distinction in literature, as well as show that TGF-β1 signaling in 3D plays an innate role in promoting skeletal muscle differentiation in monocultures of myogenic cells.

The tissue-engineered TGF-β1-treated myoblast and myofibroblast co-culture model is a promising candidate for therapeutic treatment of volumetric muscle loss and can be employed as a screening tool for pharmaceutical treatments that support muscle regeneration. Additionally, this model can be used to improve product quality of *in vitro* meat, which has experienced accelerated interest in recent years (*Langelaana et al., 2010*; *Post, 2012*). Because of growing awareness of the livestock sector's contribution to

worsening climate change, deforestation of the rain forests, reductions in biodiversity, environmental degradation (*Steinfeld et al., 2006*), and increased bacterial resistance to antibiotics (*Frieden, 2013*), devising a sustainable solution for biomanufactured meat production is imperative. *In vitro* meat is being recognized as a sustainable alternative to traditional factory farming as it uses less food, water, and land resources; produces less greenhouse gases; requires less energy for production (*Tuomisto & de Mattos, 2011*); and does not require systemic overuse of antibiotics. Since meat quality is dependent, in part, on sarcomere development, muscle fiber type, and connective tissue content (*Joo et al., 2013*), including TGF-β1 as a cell culture media supplement and co-culturing muscle cells with myofibroblasts can produce meat with flavor and texture increasingly similar to that derived from livestock.

Although our self-assembled CoTGFβ model of skeletal muscle achieved structural similarity to muscle fascicles, utilizing this self-assembly strategy for scale up of tissue engineering is not practical. Here we have shown that the CoTGFβ technique can be used in hydrogel-based systems, which possess flexibility in design parameters to incorporate vasculature (*Koffler et al., 2011*; *Levenberg et al., 2005*). Additionally, electrical stimulation of muscle fibers further matures *in vitro* skeletal muscle models by accelerating sarcomere development (*Langelaan et al., 2011*) and increasing contraction force (*Fuoco et al., 2015*; *Ito et al., 2014*). Accordingly, applying vascularization and electrical stimulation regimens to TGF-β1-treated myofibroblast and myoblast co-cultures shows promise to improve engineered skeletal muscle's recapitulation of native muscle.

## ACKNOWLEDGEMENTS

The authors thank Dr. Marsha Rolle at Worcester Polytechnic Institute for graciously providing a PDMS mold from which agarose molds were based. We additionally thank faculty and staff at Northeast Ohio Medical University: Dr. William Chilian for supplying C2C12 cells for experiments, Dr. Zhenyu Jia for his assistance with the statistics used in this study, and Dr. Marc Penn and technician Matthew Kiedrowski for use of the Penn lab's Olympus Multiphoton Microscope.

### Funding

This work was supported by Kent State University and the "Bioreactor Design for Cultured Pork" grant awarded to Jess Krieger by New Harvest, a 501(c)(3) non-profit research institute. The funders had no role in study design, data collection and analysis, decision to publish, or preparation of the manuscript.

### Grant Disclosures

The following grant information was disclosed by the authors:
Kent State University
"Bioreactor Design for Cultured Pork".

## Competing Interests

The authors declare there are no competing interests.

## Author Contributions

- Jessica Krieger conceived and designed the experiments, performed the experiments, analyzed the data, contributed reagents/materials/analysis tools, prepared figures and/or tables, authored or reviewed drafts of the paper, approved the final draft.
- Byung-Wook Park performed the experiments, analyzed the data, contributed reagents/materials/analysis tools, authored or reviewed drafts of the paper.
- Christopher R. Lambert performed the experiments, contributed reagents/materials/-analysis tools.
- Christopher Malcuit conceived and designed the experiments, analyzed the data, contributed reagents/materials/analysis tools, authored or reviewed drafts of the paper.

## Data Availability

The raw data are provided in the Supplemental Files.

## Supplemental Information

Supplemental information for this article can be found online at http://dx.doi.org/10.7717/peerj.4939#supplemental-information.

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
