# Peer review of "D skeletal muscle fascicle engineering is improved with TGF-β1 treatment of myogenic cells and their co-culture with myofibroblasts"

_PeerJ, doi:10.7717/peerj.4939_

## Round 0.1 · original submission · Major Revisions

Relevant previous works on the topic should be carefully cited (as suggested by the reviewer) and modify the title according to the reviewer suggestion, if appropriate.

·

Basic reporting

Overall, the manuscript was well written. The organization was logical and statements supported by data or the literature. The authors should have cited Am J Physiol Cell Physiol 2001;280:C288-95 since this work describes the importance of the addition of fibroblasts to for functional myotubes in 3D scaffold-free cultures.

Line 154. The reference Baaijens et. al was not provided. Note the phrase is "et al." not "et. al".

Figure 1 should indicate when TGFbeta was added relative to the start of differentiation. Also, clarify in the methods section whether the time of TGFbeta addition was the same in 2D and 3D culture.

Experimental design

The experimental design largely follows that of of other studies and the hypothesis is not novel. The methods are generally well described and can be reproduced with one comment noted below. As noted above Dennis et al. (2001) showed the importance of adding fibroblasts to enhance myotube formation and increase contractile force generation in 3D scaffold-free muscle cultures. The studies with TGFbeta largely replicate and validate the work of Weist et al. Consequently, the investigators do not provide a novel hypothesis or indicate what new information will be sought.

Figure 1 should indicate when TGFbeta was added relative to the start of differentiation. Also, clarify in the methods section whether the time of TGFbeta addition was the same in 2D and 3D culture.

Validity of the findings

Overall, this study was technically well done but adds little to our understanding of the role of TGFbeta on the function of 3D muscle cultures. The effect of TGFbeta on myofibroblasts in 3D is rather modest given the large number of nuclei shown in the supplemental figure and does not appear to effect function or the ability to maintain 3D cultures for 24 hours or longer. The effect of TGFbeta on myoblasts is significant and agrees with the earlier work of Weist et al. Further, Dennis et al. (Am J Physiol Cell Physiol 2001;280:C288-95) have previously shown the benefits of adding fibroblasts to scaffold-free 3D cultures, although the current study provides some additional details of the impact of fibroblasts to myoblast 3D cultures. The new information is the effect of the various treatments on the Young's modulus, the dimensions of the constructs and the organization of myotubes with added fibroblasts.

The authors argue that removal of the 3D cultures from the posts is analogous to the slack test applied to muscle fibers to measure the zero force velocity. Removal of the muscle rings from the posts does eliminate tension arising from forces required to maintain the diameter of the post. However, since the dimension of the ring depend on conditions, a tension remains after removal of the post due to the passive forces arising from differences in constructs. Cutting the ring would remove any internal force and replicate the slack test.

The enhancement of myofiber formation is consistent with the work of Weist et al., although they observed more myofibers present in the absence of TGFbeta, possibly a reflection of the use of primary cells.

The manuscript would be stronger if new mechanistic of functional insights were obtained. As is, the study is largely duplicative and does not increase our understanding about the role of TGFbeta in 3D muscle cultures.

·

Basic reporting

no comment

Experimental design

1) The title of the article states ‘Skeletal muscle Facile engineering is improved by co-culture with myofibroblasts and TGF-β1’. The term " co-culture" can be used only if two cell types are cultured together and do not apply to co-culture with growth factors or supplements. Hence the title should be restructured to include another cell type or changed to supplementation of TGF-β1 and not co-culture. Also, it should also include that it specifically in 3D culture and not just co-culture.

2) The authors should explain in detail about their self-assembled agarose tissue culture since it is not mentioned in the methods section. They mentioned that the custom agarose rings with central mold, but it is not clear why they chose that design. I strongly suggest them to provide a clear picture of the mold and explain the structure in detail.

3) Also, authors have mentioned that the cell seeding density was 350k cells/millimeter of post diameter, but did not explain why this seeding density was chosen or is there any data that supports this cell density. The self-assembled structure design and the cell density in the structure are highly critical because this is the basis of the mechanical testing results that the authors had obtained, so this has to be explained clearly and thoroughly.

4) The authors treated the cells with 1ng/ml TGF-β1 in this study. Will the authors be able to provide data on what will be the dose-response of TGF-β1 in myotubule expression on the cells, to select the optimal amount of TGF-β1 required? Also, if this optimal amount varies between 2D and 3D cultures.

5) In 2D cultures of the fibroblasts, the authors stated that there is no difference in collagen 1 transcription level during TGF-β1 treatment but reported that increased collagen expression in immunostaining. There was no explanation given as to why they observe this or why they think that post-translation collagen expression is higher.

6) The authors did not provide any information on the size of the self-assembled agarose 3D cultures or collagen 3D hydrogels. It might be a fact that some of the downregulated genes that the authors are observing may be because the gels are diffusion limited. Live/dead staining images of the cells in the hydrogels to see the percentage of the cells survive in the hydrogel environment would be very informative. Authors should consider perfusion systems for culturing 3D tissues otherwise the cultures may be diffusion limited preventing the mass transfer of nutrients and gas exchange.

7) The authors mentioned that the expression of collagen is downregulated by fibroblasts in 3D cultures compared to 2D cultures. In fig H they show that just by the addition of fibroblast to myoblasts in 3D cultures Young's modulus of the hydrogel increases. If this indeed is a factor why does the modulus is increasing although the collagen 1 expression is down-regulated in 3D cultures? The authors should provide a reasonable explanation for this.

Validity of the findings

no comment

---

## Round 0.2 · accepted · Accept

Line 513 still has et. al, Make sure that the correct way of cite is: et al.,

# ·

Basic reporting

No comment

Experimental design

No Comment

Validity of the findings

No Comment

Additional comments

The issues I raised in my original review were adequately addressed.